# Immunogenicity of the 13-Valent Pneumococcal Conjugate Vaccine (PCV13) Followed by the 23-Valent Pneumococcal Polysaccharide Vaccine (PPSV23) in Adults with and without Immunosuppressive Therapy

**DOI:** 10.3390/vaccines10050795

**Published:** 2022-05-17

**Authors:** Hannah M. Garcia Garrido, Albert Vollaard, Geert R. D’Haens, Phyllis I. Spuls, Frederike J. Bemelman, Michael W. Tanck, Godelieve J. de Bree, Bob Meek, Martin P. Grobusch, Abraham Goorhuis

**Affiliations:** 1Amsterdam UMC, Centre of Tropical Medicine and Travel Medicine, Department of Infectious Diseases, Amsterdam Institute for Infection and Immunity, University of Amsterdam, 1105 AZ Amsterdam, The Netherlands; g.j.debree@amsterdamumc.nl (G.J.d.B.); m.p.grobusch@amsterdamumc.nl (M.P.G.); a.goorhuis@amsterdamumc.nl (A.G.); 2Center for Infectious Disease Control Netherlands, National Institute for Public Health and the Environment, 3721 MA Bilthoven, The Netherlands; albert.vollaard@rivm.nl; 3Amsterdam UMC, Department of Gastroenterology, University of Amsterdam, 1105 AZ Amsterdam, The Netherlands; g.dhaens@amsterdamumc.nl; 4Amsterdam UMC, Department of Dermatology, University of Amsterdam, 1105 AZ Amsterdam, The Netherlands; ph.i.spuls@amsterdamumc.nl; 5Amsterdam UMC, Department of Nephrology, University of Amsterdam, 1105 AZ Amsterdam, The Netherlands; f.j.bemelman@amsterdamumc.nl; 6Amsterdam UMC, Department of Epidemiology and Data Science, University of Amsterdam, 1105 AZ Amsterdam, The Netherlands; m.w.tanck@amsterdamumc.nl; 7St. Antonius Hospital, Department of Medical Microbiology and Immunology, 3435 CM Nieuwegein, The Netherlands; b.meek@antoniusziekenhuis.nl

**Keywords:** pneumococcal vaccination, immunocompromised host, transplant recipient, autoimmune disease, vaccine immunogenicity

## Abstract

Immunosuppressive therapy increases the risk of pneumococcal disease. This risk can be mitigated by pneumococcal vaccination. The objective of this study was to investigate the immunogenicity of the 13-valent pneumococcal conjugate vaccine (PCV13), followed by the 23-valent pneumococcal polysaccharide vaccine (PPSV23), in adults with and without immunosuppressive therapy. We performed a prospective cohort study among adults using conventional immunomodulators (cIM), biological immunomodulators (bIM), combination therapy, and controls during 12 months. The primary outcome was seroprotection, defined as the proportion of patients with a postimmunization IgG concentration of ≥1.3 µg/mL for at least 70% (17/24) of the serotypes of PCV13 + PPSV23. We included 214 participants. For all 24 vaccine serotypes, IgG levels increased significantly in both treatment subgroups and controls, with peak seroprotection rates of 44% (combination therapy), 58% (cIM), 57% (bIM), and 82% (controls). By month 12, seroprotection had decreased to 24%, 48%, 39%, and 63%, respectively. Although pneumococcal vaccination with PCV13 + PPSV23 was immunogenic in all treatment groups, impaired vaccination responses were observed in patients using immunosuppressive medication. Apart from the obvious recommendation to administer vaccines before such medication is started, alternative vaccination strategies, such as additional PCV13 doses or higher-valent pneumococcal vaccines, should be investigated.

## 1. Introduction

The risk of invasive pneumococcal disease (IPD) is increased in patients with autoimmune diseases (6.5-fold) and in solid organ transplant recipients (47-fold), compared with the general population [1]. Immunosuppressive therapy, the cornerstone of the treatment of these patients aiming at the suppression of autoimmunity and the prevention of graft rejection, increases the susceptibility to and severity of pneumococcal infections [2,3,4]. In addition, the response to vaccination is generally impaired [5,6]. Pneumococcal vaccination with the 13-valent pneumococcal conjugate vaccine (PCV13), followed 2 months later by the 23-valent polysaccharide vaccine (PPSV23), is currently recommended for patients with autoimmune diseases and solid organ transplant recipients in most countries [7,8]. Recall antipneumococcal responses have been demonstrated in healthy elderly when the two vaccines were used in series [9]. However, data regarding the strength and duration of seroprotection resulting from this combined vaccination in patients using immunosuppressive agents are scarce, with most studies focusing on underlying diagnoses rather than the type of immunosuppressive medication [10]. In addition, most prior studies tended to focus on the immunogenicity of PCV serotypes rather than the immunogenicity of serotypes exclusive to PPSV23 [11,12], which have become predominant in recent years, due to serotype replacement [13,14].

Therefore, the main objective of the present study was to investigate the immune response to the recommended vaccination schedule of PCV13, followed 2 months later by PPSV23, for all 24 serotypes included in PCV13 and PPSV23, in adults using immunosuppressive medication and controls during 12 months.

## 2. Materials and Methods

We performed a prospective interventional cohort study at the Amsterdam UMC, location AMC, between August 2018 and December 2021 (NL7193).

### 2.1. Study Population

The following categories of consecutive adults visiting the Amsterdam UMC Center of Tropical Medicine and Travel Medicine’s vaccination clinic to receive their routine, or travel, vaccinations were asked to participate: (1) patients using conventional immunomodulators (cIM), (2) patients using biological immunomodulators (bIM), (3) patients using combination therapy, and (4) controls using no systemic immunosuppressive treatments (Figure 1). Low-dose prednisolone <10 mg/day or <700 mg cumulative was not considered immunosuppressive [15]. If a patient switched to a different medication category within 2 weeks after enrollment, the patient was assigned to that new category for final analysis. If immunosuppressive treatment was initiated or stopped between 2 weeks after enrollment until 2 weeks after PPSV23 administration, the patient was assigned to the ‘switched group’ (Figure 1).

The following exclusion criteria were applied: previous vaccination with any PCV, vaccination with PPSV23 in the past 5 years, diagnosis of a primary immune deficiency disorder, hemophilic disorder precluding intramuscular vaccination, allergy to any of the components of the pneumococcal vaccines, pregnancy, not being able or willing to consent.

### 2.2. Sample Size

The power calculation was based on expected differences in seroprotection rates after vaccination between patients using immunosuppressive medication and controls. Based on previous studies, we expected that in our population, no more than 60% of patients using immunosuppressive medication and 90% of controls would achieve seroprotection [5]. A Fisher’s exact test with a 0.05 two-sided significance level will have 80% power to detect this difference when the sample size in each medication group is 36 (nQuery, version 7.0; Statsols, Cork, Ireland). To compensate for an expected 15% dropout and 20% medication switches, we planned to recruit at least 49 individuals per medication group.

### 2.3. Study Procedures

Participants received one dose of PCV13, followed by one dose of PPSV23, with a 2-month interval. PCV13 or Prevenar 13^®^ (Pfizer, New York, NY, USA) includes purified capsular polysaccharide of 13 serotypes of *Streptococcus pneumoniae* (1, 3, 4, 5, 6A, 6B, 7F, 9V, 14, 19A, 19F, 18C, and 23F) conjugated to a nontoxic variant of diphtheria toxin known as CRM197. PPSV23 or Pneumovax 23^®^ (Merck Sharp & Dohme, Kenilworth, NJ, USA) contains purified capsular polysaccharide of 23 serotypes (1, 2, 3, 4, 5, 6B, 7F, 8, 9N, 9V, 10A, 11A, 12F, 14, 15B, 17F, 18C, 19F, 19A, 20, 22F, 23F, and 33F). Some participants concomitantly received other recommended vaccinations, such as hepatitis B, influenza, or travel vaccines. All vaccines were administered intramuscularly. Prior to the first vaccination (T0), baseline clinical and demographical data were collected. Serum samples were collected at baseline and at 2, 4, 6, and 12 months after enrollment, and were frozen at −80 °C until further analysis. Serotype-specific pneumococcal IgG serum concentrations were measured using a 26-plex multiplex immunoassay as described previously [16]. After each vaccination, participants were asked to record adverse events (AEs) through an online questionnaire. Serious adverse events (SAEs) were recorded throughout the study period.

### 2.4. Outcomes and Analysis

The primary outcome of this study was the overall seroprotection rate 2 months after the full vaccination schedule (month 4), defined as the proportion of patients with a postimmunization IgG concentration of ≥1.3 µg/mL for at least 70% (17/24) of the serotypes of PCV13 + PPSV23 in the different medication groups [17].

Secondary outcomes were: (1) seroprotection rates for each individual serotype and for the group of serotypes in PCV13 (1, 3, 4, 5, 6A, 6B, 7F, 9V, 14, 19A, 19F, 18C, and 23F) and PPVS23-unique serotypes (2, 8, 9N, 10A, 11A, 12F, 15B, 17F, 20, 22F, 33F); (2) serotype-specific geometric mean concentrations (GMC) of IgG for all 24 serotypes and geometric mean fold rises (GMFRs) compared with baseline; (3) clinical and laboratory predictors for the primary outcome; and (4) dynamics of seroprotection rates and GMCs over time. Sensitivity analyses were performed to see whether the results would change if patients using anti-CD20 therapy were excluded from the analysis, as this therapy is known to impair immune responses more profoundly.

For all analyses, we used SPSS (IBM, Chicago, IL, USA) version 25.0 or higher. Antibody concentrations were analyzed on a log-transformed scale and presented as GMC. To detect changes in serotype-specific protection rates and GMC of IgG over time, generalized linear mixed models (GLMM, covariance structure first order autoregression) including the variables ‘time point’ and ‘immunosuppressive medication group at baseline’ were used. To identify predictors for the primary outcome, we performed a multivariable logistic regression analysis including sex and age as fixed variables and using stepwise backward selection based on likelihood ratio and *p* < 0.05 for the following predefined variables: body mass index, Charlson comorbidity index, smoking (current smoker yes/no), alcohol use (>7/week, yes/no), illicit drug use (any type, yes/no), type of immunosuppressive medication (any cIM, high-dose steroids, methotrexate, thiopurine, calcineurin inhibitor, mycophenolate mofetil, any bIM, TNF-alpha inhibitor, bIM/non-TNF-alpha, anti-CD20 therapy, ustekinumab), number of immunosuppressive drugs, underlying diagnosis (rheumatoid arthritis, Crohn’s disease, ulcerative colitis, psoriasis, solid organ transplantation), impaired kidney function, use of mucosal agents (5-aminosalicylates/vedolizumab), low-dose prednisolone. We used a two-sided alpha level of 0.05 for significance of statistical tests. Missing data were excluded from analysis. All analyses were performed per protocol: all participants who had pre- and postvaccination serum samples collected were included in the final analysis.

### 2.5. Ethical Considerations

The study was conducted according to the principles of the Declaration of Helsinki and in accordance with the Medical Research Involving Human Subjects Act (WMO). The research ethics committee of the Amsterdam UMC, location AMC, in 2018, provided ethical clearance (NL65687.018.18). All participants provided written informed consent for the study.

## 3. Results

Between August 2018 and December 2020, 233 individuals were recruited, of which 214 were included in the final analysis, and 200 completed the study (Figure 2, Table 1). There were several differences across the five medication groups, including age distribution, underlying diagnosis, Charlson comorbidity index, kidney function, and the type and number of immunosuppressive agents (Table 1).

### 3.1. Seroprotection Rates

Overall seroprotection rates at month 4 ranged between 44% (combination therapy group) and 82% (control group). Seroprotection rates in the cIM, bIM, and switched groups (58%, 57%, and 60%, respectively) did not statistically differ from each other (Figure 3A, Table 2). Results were similar when only PCV13- or PPSV23-exclusive serotypes were considered, except that seroprotection was significantly higher for PCV13 serotypes in the switched group (60%) compared with the other medication groups (46%–53%) (Figure 3B,C, Table 2). Results were also similar when participants using anti-CD20 therapy were excluded from the analysis (Appendix A). The protection for PPSV23-unique serotypes was slightly higher compared with PCV13 serotypes for the bIM, cIM, and combination therapy groups, but this difference was not significant (data not shown).

The protection rate was highly serotype specific and ranged from 15% to 97% for the individual serotypes, with poor immunogenicity for serotypes 3 and 12F in all participants and for serotypes 4 and 5 in patients using any category of immunosuppressive drugs (Appendix A). Individual protection rates increased significantly over time for all vaccine serotypes (Appendix A).

### 3.2. Predictors of Seroconversion

In univariable analysis, impaired kidney function, greater number of drugs at baseline, and the use of methotrexate were significantly associated with impaired seroprotection at month 4 (Table 3). In the multivariable model, significant negative predictors of the primary outcome at month 4 were: solid organ transplantation (adjusted odds ratio (aOR), 0.40; 95% confidence interval (CI), 0.17–0.97), use of methotrexate at baseline (aOR, 0.38; 95% CI, 0.17–0.81), and anti-CD20 therapy (aOR, 0.09; 95% CI, 0.01–0.81) (Table 3).

When considering all time points, the immunosuppressive treatment group had a significant impact on protection rates for the following vaccine serotypes: 4, 5, 9V, 12F, 19A, 20, 23F (Appendix A). Both time point and medication groups were statistically significant predictors for overall response rate during 12 months (Appendix A).

### 3.3. Waning Immunity

For all time points, the overall seroprotection rate was higher compared with baseline; however, at month 12, protection rates dropped to 63% in controls and to 28% in patients using combination therapy and ranged between 39% and 48% in the cIM, bIM, and switched groups (Table 2 and Appendix A).

For all vaccine serotypes, GMCs increased significantly over time. Peak concentrations for PCV13 serotypes were achieved at T2 or T4 and exceeded the protective cut-off, except for serotype 3. Peak GMFRs ranged from 3.9 to 31. Between T4 and T12, GMCs declined but remained significantly higher compared with baseline (GMFRs between 2.4 and 14). For several serotypes, GMCs dropped below the protective cut-off by the end of the follow-up for at least one of the study groups (serotypes 1,4, 5, 6B, 9V, 23F) (Figure 4A, Appendix A).

For PPSV23-unique serotypes, peak concentrations were achieved at T4 and exceeded the protective cut-off for all serotypes except 12F, with peak GMFRs ranging from 4.7 to 32. Between T4 and T12, GMCs declined but remained higher compared with baseline, with GMFRs between 2.9 and 12. At month 12, GMCs remained above the protective level for all PPSV23-unique serotypes in all treatment groups except for serotypes 10A and 22F (Figure 4B, Appendix A).

### 3.4. Safety

No cases of IPD or pneumococcal pneumonia occurred during the study period. No SAEs related to vaccination occurred (Appendix A). Vaccination was well tolerated as only mild side effects were reported after vaccination in 38% of the participants. Overall, 7% of the participants reported exacerbations of their underlying autoimmune diseases during the study period of 12 months, but these were not temporally associated with vaccination.

## 4. Discussion

In this prospective study, the immunogenicity of the recommended pneumococcal vaccination schedule combining PCV13 and PPSV23 among patients using immunosuppressive medication was investigated. Vaccination was safe and immunogenic with seroprotection rates and GMCs increasing significantly for all pneumococcal serotypes in both treatment groups and controls. However, in all treatment groups, the peak seroprotection rates following vaccination (44%–60%) were impaired compared with controls (82%), and GMCs were lower for 12/24 (50%) of vaccine serotypes. Our data are in line with a prior study investigating the recommended vaccination schedule among patients with inflammatory bowel disease (IBD: protection of 81% in controls and 52%–63% in medication groups) [5] and a small study in rheumatoid arthritis patients (protection of 55%–63%). Both studies used the same correlate of protection as the present study. Much higher protection rates (>87%) are reported in studies using the 0.35 µg/mL WHO cut-off [11,12]. However, data from clinical studies have pointed out that the WHO cut-off is likely too low, leading to an overestimation of vaccine efficacy against IPD, especially in adults [16,17,18,19,20].

### 4.1. The Effect of Different Immunosuppressive Agents on Seroprotection

We found that patients on combination therapy had the lowest protection rate (44%), while protection rates did not significantly differ between patients on cIM and bIM monotherapy. This has been reported in previous studies among patients with IBD [5,21]. The use of methotrexate was an independent predictor of poor seroprotection after vaccination in this study, corresponding with findings from prior research [10,22,23]. This may be due to the broad and less specific mechanism of action of methotrexate, limiting the expansion of all lymphocyte populations by inhibiting folic acid metabolism [24]. In line with previous literature, the use of anti-CD20 therapy, resulting in B-cell depletion, was evidently associated with a very poor humoral response to vaccination (6%) [10,12]. The history of pneumococcal vaccination has been found to be protective against severe bacterial infections in patients using rituximab (OR, 0.11; 95% CI, 0.03–0.41), emphasizing the importance of vaccination prior to initiation of anti-CD20 therapy [25]. Last, the poor seroprotection in solid organ transplant recipients likely has a multifactorial cause, such as the use of triple immunosuppressive therapy, higher rates of comorbidities, and an impaired kidney function, the latter of which increases both the risk of pneumococcal infections and nonresponse to vaccination [4,26]. Vaccination of patients on the waiting list for transplantation should thus be recommended.

Interestingly, we found significantly higher short-term responses for PCV13 serotypes for the switched group compared with other treatment groups, although responses were still lower than for controls during the entire study. This is relevant, as there is not always enough time to postpone immunosuppressive treatment until after the entire pneumococcal vaccination schedule has been completed. Based on our results, we would then recommend to only administer PPSV23 instead of the combined schedule.

### 4.2. Different Pneumococcal Vaccination Schedules

PCV induces immunological memory, contrary to PPSV23 that only elicits a (T-cell-independent) plasma cell response. However, contrary to expectations, prior studies have demonstrated that the immunogenicity of a single PCV dose in patients using immunosuppressive drugs is relatively similar to that of PPSV23 [5,26]. This may be due to immunosuppressive medication impairing T-helper cell responses needed for an adequate response to PCV. The value of PCV in immunocompromised patients could be to overcome initial low responses by the administration of multiple doses of PCV to enhance immune maturation and development of memory B cells [16,27,28]. It would be of interest to investigate whether an additional PCV13 dose improves the primary response to pneumococcal vaccination in solid organ transplant recipients and patients using combination therapy or methotrexate, as these were at risk of nonresponse to vaccination in the present study. Considering the increase in incidence of non-PCV13 serotypes as a cause of IPD, this strategy may be expanded to include a novel 20-valent pneumococcal conjugate vaccine that was recently approved by the Food and Drug Administration. Another 24-valent candidate vaccine is under investigation. Future studies should focus on the immunogenicity, duration of seroprotection, and cost-effectiveness of these novel vaccines as a single dose, in multiple doses, or in series with PPSV23 among patients using immunosuppressive drugs, as compared with PCV13/PPSV23 and PPSV23 alone.

The benefits of the PCV13/PPSV23 combined vaccination schedule compared with PPSV23 alone are controversial. In a randomized controlled trial directly comparing PPSV23 with PCV7/PPSV23 in liver transplant recipients, priming with PCV7 did not enhance the immunogenicity for seven serotypes contained in both vaccines [11]. On the other hand, a retrospective clinical study among U.S. veterans with inflammatory bowel disease demonstrated a protective effect against pneumococcal disease of PCV+PPSV23 and PCV alone, but not of PPSV23 [29]. In addition, in the present study, we also report higher seroprotection rates compared with studies investigating single PCV/PPSV23 vaccination, despite hyporesponsiveness (i.e., the absence of an IgG increase following PPSV23 vaccination) for 7/13 PCV13 serotypes. A meta-analysis including 764 patients with autoimmune diseases using immunosuppressive drugs reported a response rate (≥2-fold increase for serotype 6B + 23F) of 26% for PCV and 37% for PPSV23 [10]. When using the same definition of protection in a post hoc analysis, we found a protection rate of 77%–82%. Taken together, clinical and seroprotection data point towards superiority of the combined schedule.

### 4.3. Duration of Protection

The duration of protection is lower in patients using immunosuppressive drugs versus controls, mainly due to a lower primary response to vaccination, as waning of protection occurred at the same rate in all medication groups and controls (10%–19% decrease in protection by month 12). Further decay of protection (3 years after initial vaccination) is currently being investigated in the same cohort. Data on duration of protection after pneumococcal vaccination among patients using immunosuppressive drugs are scarce. One study in organ transplant recipients reported a decrease in protection of 21% over 3 years (n = 47), without differences between PCV and PPSV23 [30]. Another study in rheumatoid arthritis patients reported that within 2 years after PCV13 + PPSV23, neutralizing antibody titers had decreased significantly and even below baseline levels [31]. Altogether, evidence suggests that seroprotection following vaccination may not last for 5 years, and that a single dose of PCV in series with PPSV23 may not improve the duration of protection. If the combined vaccination schedule were to be maintained, accelerating the PPSV23 booster vaccination to 3 instead of 5 years could be considered, especially for patients with predictors for nonresponse at the time of vaccination, such as solid organ transplant recipients. Alternatively, strategies to augment the primary response to vaccination, such as additional PCV doses, should be investigated. Lastly, monitoring of pneumococcal IgG levels from 1 year after vaccination and antibody-directed booster vaccinations may be an option, although this might not be feasible in all settings.

A limitation of this study was that it was not powered to assess clinical endpoints. In addition, correlates of protection for IPD remain controversial. The correlate of protection used in this study was rather conservative and may have led to an underestimation of protection. We therefore also analyzed and reported GMC data.

## 5. Conclusions

Pneumococcal vaccination with PCV13 followed by PPSV23 was safe and immunogenic in patients using immunosuppressive medication and controls, with seroprotection rates and GMCs increasing significantly for all vaccine serotypes. However, as seroprotection was significantly impaired in patients using immunosuppressive medication, vaccination prior to the start of immunosuppressive treatment or organ transplantation is highly recommended. In addition, due to waning immunity, an earlier-than-5-year PPSV23 booster should be considered. Future studies must elucidate whether additional doses of PCV13 or higher-valent pneumococcal vaccines are beneficial for patients at risk of poor protection after vaccination, such as solid organ transplant recipients.

## Figures and Tables

**Figure 1 vaccines-10-00795-f001:**
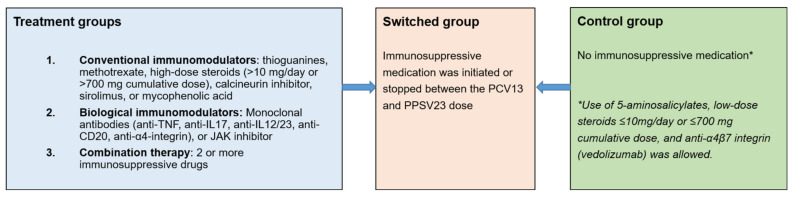
Study groups. TNF = tumor necrosis factor, IL = interleukin, JAK = Janus kinase, PCV13 = 13-valent pneumococcal conjugate vaccine, PPSV23 = 23-valent pneumococcal polysaccharide vaccine.

**Figure 2 vaccines-10-00795-f002:**
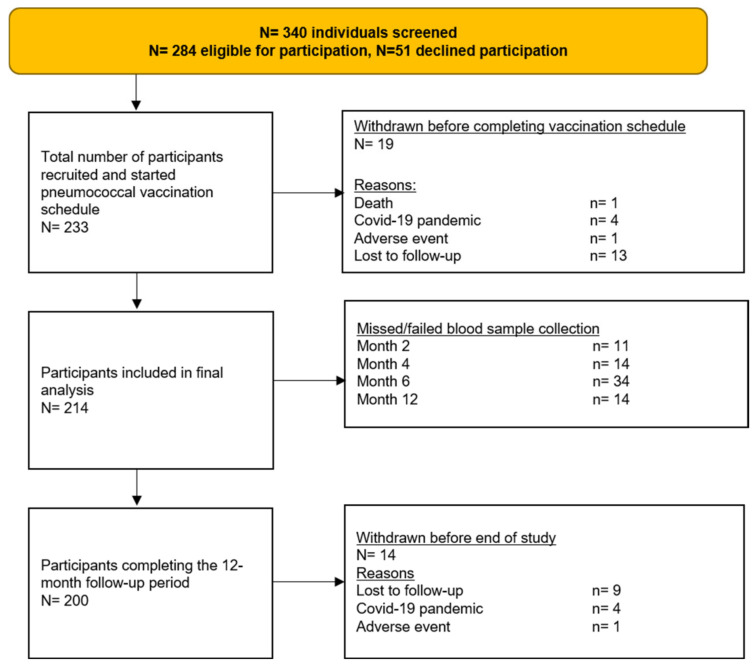
Study flow diagram.

**Figure 3 vaccines-10-00795-f003:**
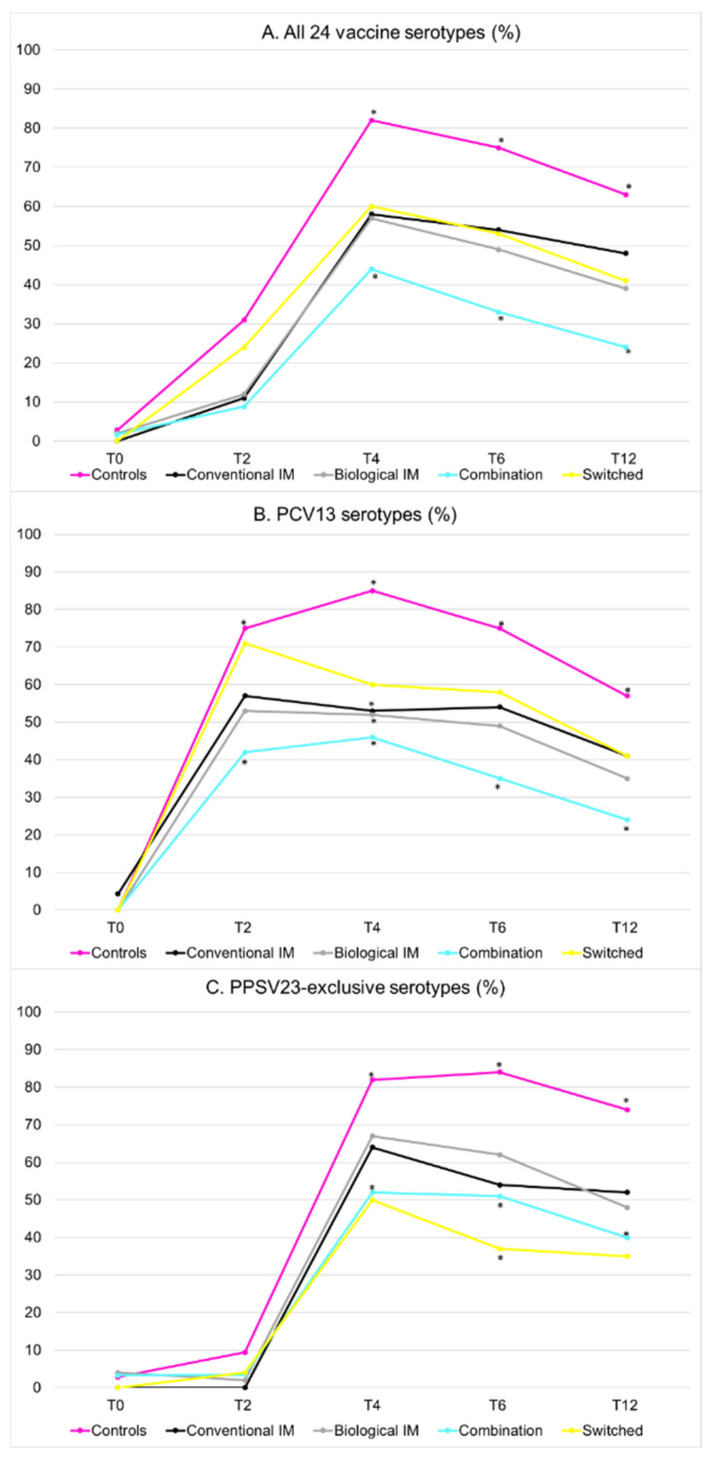
(**A**) Seroprotection rates over time for all 24 vaccine serotypes; (**B**) Seroprotection rates over time for serotypes included in PCV13; (**C**) Seroprotection rates over time for serotypes exclusive to PPSV23. IM = immunomodulator. T = time point in months from enrollment. * Highlights seroprotection rates that differ significantly from the other groups at that time point.

**Figure 4 vaccines-10-00795-f004:**
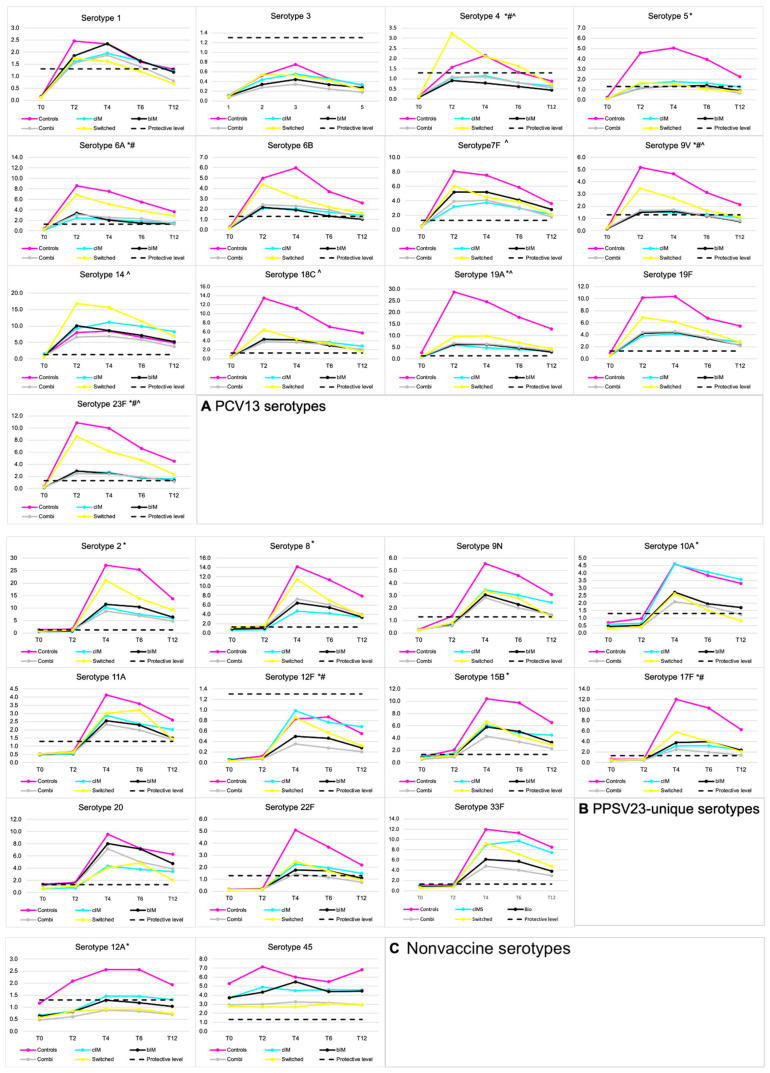
Serotype-specific GMCs over time for serotypes present in PCV13 (**A**), PPSV23 only (**B**), and two nonvaccine control serotypes (**C**). * Indicates statistically significant effect (*p* < 0.05) of treatment groups on geometric mean concentrations. # Indicates significant interaction effect of time point * treatment group on geometric mean concentration. ^ Indicates the presence of PPSV23 hyporesponse (absence of significant increase following PPSV23). cIM = conventional immunomodulator. bIM = biological immunomodulator. PCV13 = 13-valent pneumococcal conjugate vaccine. PPSV23 = 23-valent pneumococcal polysaccharide vaccine. T = time point in months from enrollment.

**Table 1 vaccines-10-00795-t001:** Baseline characteristics.

	Total Cohort *N* = 214	Groups Based on Use of Immunosuppressive Medication					
		Conventional Immunomodulators *N* = 47	Biological immunomodulators *N* = 50	Combination Therapy *N* = 60	Switched *N* = 21 ^1^	Controls *N* = 36	*p*-Value across Groups
Males *n* (%)	100 (47)	20 (43)	26 (52)	32 (53)	11 (52)	11 (31)	0.20
Age, median (IQR ^2^)	41 (26)	47 (30)	40 (26)	41 (20)	30 (23)	48 (24)	0.10
Age group 18–49 *n* (%)	139 (65)	26 (55)	33 (66)	45 (75)	17 (81)	18 (50)	**0.03**
Age group 50–70 *n* (%)	75 (35)	21 (45)	17 (34)	15 (25)	4 (19)	18 (50)	
Body mass index, median (IQR)	24 (6)	24 (5)	25 (6)	24 (7)	25 (6)	23 (7)	0.42
Current smoker *n* (%)	32 (15)	7 (15)	7 (14)	6 (10)	5 (24)	7 (19)	0.55
Previous smoker *n* (%)	54 (25)	10 (21)	14 (28)	11 (18)	7 (33)	12 (33)	0.39
Alcohol use >7/week *N* (%)	41 (19)	7 (15)	6 (12)	16 (17)	4 (19)	8 (22)	0.33
Drug use (yes/no) *n* (%)	21 (9.8)	5 (11)	2 (4.0)	6 (10)	5 (24)	3 (8.3)	0.15
**Underlying disease**							
Crohn’s disease *n* (%)	48 (22)	8 (17)	9 (18)	9 (15)	9 (43)	13 (36)	**0.02**
Ulcerative colitis *n* (%)	32 (15)	6 (13)	9 (18)	7 (11)	3 (14)	7 (19)	0.8
Rheumatoid arthritis *n* (%)	31 (14)	14 (30)	6 (12)	8 (13)	2 (9.5)	1 (2.8)	**0.01**
Psoriasis/psoriatic arthritis *n* (%)	32 (15)	6 (13)	14 (28)	5 (8.3)	4 (19)	3 (8.3)	**0.03**
Spondylitis ankylopoietica *n* (%)	7 (3.3)	0 (0)	4 (8.0)	0 (0)	2 (9.5)	1 (2.8)	**0.04**
Neurological autoimmune disease *n* (%)	7 (3.3)	1 (2.1)	5 (10)	1 (1.7)	0 (0)	0 (0)	**0.05**
Other autoinflammatory disease ^3^ *n* (%)	21 (9.8)	10 (21)	3 (6)	3 (5)	1 (4.8)	4 (11)	**0.04**
Solid organ transplant recipient ^4^ *n* (%)	29 (14)	2 (4.3)	0 (0)	27 (45)	0 (0)	0 (0)	**<0.01**
Comorbidity score (Charlson comorbidity index) median (IQR)	1 (1)	1 (2)	1 (1)	2 (2)	1 (2]	1.5 (1)	**0.03**
Pulmonary disease in medical history *n* (%)	15 (7.0)	4 (8.5)	4 (8.0)	3 (5.0)	1 (4.8)	4 (11)	0.79
Impaired kidney function (eGFR ^5^ < 60) *n* (%)	24 (11)	3 (6.4)	2 (4.0)	17 (28)	0 (0)	2 (5.6)	**<0.01**
**Number of drugs at baseline**							
One drug *n* (%)	98 (46)	47 (100)	50 (100)	0 (0)	1 (4.8)	0 (0)	**<0.01**
Two drugs *n* (%)	39 (18)	0 (0)	0 (0)	39 (65)	0 (0)	0 (0)	
Three drugs *n* (%)	21 (9.8)	0 (0)	0 (0)	21 (35)	0 (0)	0 (0)	
**Conventional immunomodulator *n* (%)**	92 (43)	47 (100)	0 (0)	45 (75)	0 (0)	0 (0)	**<0.01**
Prednisolone (>10 mg/day or 700 mg cumulative) (%)	38 (18)	2 (4.3)	0 (0)	36 (60)	0 (0)	0 (0)	**<0.01**
Thiopurine *n* (%)	28 (13)	15 (32)	0(0)	13 (22)	0 (0)	0 (0)	**<0.01**
Methotrexate (7.5–30 mg/week) *n* (%)	37 (17)	21 (45)	0 (0)	16 (27)	0 (0)	0 (0)	**<0.01**
Calcineurin inhibitor *n* (%)	25 (12)	2 (4.3)	0 (0)	23 (38)	0 (0)	0 (0)	**<0.01**
Mycophenolate mofetil *n* (%)	27 (13)	5 (11)	0 (0)	22 (37)	0 (0)	0 (0)	**<0.01**
Other *n* (%) ^6^	4 (1.9)	2 (4.3)	0 (0)	2 (3.3)	0 (0)	0 (0)	0.38
**Biological immunomodulator *n* (%)**	79 (37)	0 (0)	50 (100)	28 (47)	1 (4.8)	0 (0)	**<0.01**
**TNF-alpha inhibitor at baseline *n* (%)**	60 (28)	0 (0)	32 (64)	24 (40)	4 (19)	0 (0)	**<0.01**
Etanercept *n* (%)	7 (3.3)	0 (0)	5 (10)	2 (3.3)	0 (0)	0 (0)	**0.03**
Infliximab *n* (%)	22 (10)	0 (0)	10 (20)	12 (20)	0 (0)	0 (0)	**<0.01**
Adalimumab *n* (%)	22 (10)	0 (0)	15 (30)	7 (12)	0 (0)	0 (0)	**<0.01**
Certolizumab pegol *n* (%)	3 (1.4)	0 (0)	1 (2.0)	2 (3.3)	0 (0)	0 (0)	0.52
Golimumab *n* (%)	2 (0.9)	0 (0)	1 (2.0)	1 (1.7)	0 (0)	2 (0.9)	0.75
**Other biological immunomodulators (non-TNF-alpha inhibitor) at baseline *n* (%)**	24 (11)	0 (0)	18 (36)	5 (8.3)	1 (4.8)	0 (0)	**<0.01**
Ustekinumab (anti-IL-12/23) *n* (%)	7 (3.3)	0 (0)	4 (8.0)	3 (5.0)	0 (0)	0 (0)	0.11
Rituximab/ocrelizumab (anti-CD20) *n* (%)/mean time since last dose in weeks (SD)	8 (3.7)/11 (6)	0 (0)	6 (12)/13 (4.8)	2 (3.3)/4 (2.8)	0 (0)	0 (0)	**<0.01**
Tofacitinib (JAK 1/3 inhibitor) *n* (%)	3 (1.4)	0 (0)	3 (6.0)	0 (0)	0 (0)	0 (0)	**0.04**
Secukinumab (anti IL-17A) *n* (%)	2 (0.9)	0 (0)	1 (2.0)	0 (0)	1 (4.8)	0 (0)	0.26
Other ^7^ *n* (%)	3 (1.4)	0 (0)	3 (6.0)	0 (0)	0(0)	0 (0)	**0.04**
**Luminal agents**							
Vedolizumab (α4β7-integrin) *n* (%)	10 (4.7)	0 (0)	1 (2.0)	1 (1.7)	1 (4.8)	7 (19.4)	**<0.01**
5-Aminosalicylates *n* (%)	15 (7.0)	6 (13)	4 (8.0)	2 (3.3)	1 (4.8)	2 (5.6)	0.41
Low-dose prednisolone *n* (%)	15 (7.0)	5 (11)	3 (6.0)	1 (1.7)	2 (9.5)	4 (11)	0.31

^1^*n* = 14 of the patients in the switched group did not use any medication at baseline and started medication between the two vaccine doses. *N* = 7 used medication at baseline but stopped between the two vaccine doses. ^2^ Interquartile range. ^3^ Acne ectopica, antisynthetase syndrome, eczema (*n* = 2), autoimmune hepatitis (*n* = 2), Behcet disease, takayasu vasculitis, Sjögren disease (*n* = 2), membranous glomerulonephritis, mixed connective tissue disease, sarcoidosis (*n* = 3), SLE (*n* = 3), unspecified systemic inflammatory disease (*n* = 2), uveitis. ^4^
*n* = 28 renal transplant recipients, *n* = 1 liver transplant recipient. ^5^ Estimated glomerular filtration rate. ^6^ Dimethylfumaric acid (*n* = 1), sirolimus (*n* = 1), cyclophosphamide (*n* = 1), leflunomide (*n* = 1). ^7^ Dupilumab (*n* = 1), ixekizumab (*n* = 1), natalizumab (*n* = 1). Key comparison group are in bold font.

**Table 2 vaccines-10-00795-t002:** Seroprotection rates.

All 24 Serotypes	T0 ^1^	T2	T4	T6	T12
Controls	1/36 (2.8)	10/32 (31)	28/34 (82) ^a^	24/32 (75) ^a^	22/35 (63) ^a^
cIM ^2^	0/47 (0)	5/44 (11)	26/45 (58)	21/39 (54)	21/44 (48)
bIM ^3^	1/50 (2.0)	6/49 (12)	26/46 (57)	19/39 (49)	18/46 (39)
Combination	1/60 (1.7)	5/57 (8.8)	24/54 (44) ^b^	17/51 (33) ^b^	15/58 (24) ^b^
Switched	0/21 (0)	5/21 (24)	12/20 (60)	10/19 (53)	7/17 (41)
*p*-Value across groups	0.81	0.04	**0.01**	**0.01**	**0.01**
**PCV13 ^4^ serotypes**	**T0**	**T2**	**T4**	**T6**	**T12**
Controls	0/36 (0)	24/32 (75) ^a^	29/34 (85) ^a^	24/32 (75) ^a^	20/35 (57) ^a^
cIM	2/47 (4.3)	25/44 (57)	24/45 (53) ^b^	21/39 (54)	18/44 (41)
bIM	0/50 (0)	26/49 (53)	24/46 (52) ^b^	19/39 (49)	16/46 (35)
Combination	3/60 (5.0)	24/57 (42) ^b^	25/54 (46) ^b^	18/51 (35) ^b^	14/58 (24) ^b^
Switched	0/21 (0)	15/21 (71)	12/20 (60)	11/19 (58)	7/17 (41)
*p*-v=Value across groups	0.27	**0.02**	**0.01**	**0.01**	**0.03**
**PPSV23 ^5^-exclusive**	**T0**	**T2**	**T4**	**T6**	**T12**
Controls	1/36 (2.8)	3/32 (9.4)	28/34 (82) ^a^	27/32 (84) ^a^	26/35 (74) ^a^
cIM	0/47 (0)	0/44 (0)	29/45 (64)	21/39 (54)	23/44 (52)
bIM	2/50 (4.0)	1/49 (2.0)	31/46 (67)	24/39 (62)	22/46 (48)
Combination	2/60 (3.3)	2/57 (3.5)	28/54 (52) ^b^	26/51 (51) ^b^	23/58 (40) ^b^
Switched	0/21 (0)	1/21 (4.8)	10/20 (50)	7/19 (37) ^b^	6/17 (35)
*p*-Value across groups	0.64	0.25	**0.04**	**0.01**	**0.02**

^1^ T = time point in months from enrollment. ^2^ Conventional immunomodulator. ^3^ Biological immunomodulator. ^4^ 13-valent pneumococcal conjugate vaccine. ^5^ 23-valent pneumococcal polysaccharide vaccine. ^a^ Proportions significantly higher compared with proportions in the same row; ^b^ proportions significantly lower compared with proportions in the same row. * Proportions with the same superscripted letter or no superscripted letter do not significantly differ from each other (adjusted for multiple hypothesis testing with Bonferroni correction). Bold *p*-values are significant after adjusted for Bonferroni correction. Key comparison group are in bold font.

**Table 3 vaccines-10-00795-t003:** Seroprotection rates and predictors for overall seroprotection 4 months after enrollment (T4).

	Overall Seroprotection Rate (%)	Raw Odds Ratio (95% CI ^1^)	Adjusted Odds Ratio (95% CI)
Males	55	ref	ref
Females	61	1.3 (0.74–2.3)	1.5 (0.80–2.7)
**Age**	NA ^2^	0.98 (0.96–1.0)	0.99 (0.97–1.0)
Age group 18–49	61	ref	NIM ^3^
Age group 50–70	54	0.75 (0.41–1.3)	
BMI ^4^	NA	1.0 (0.95–1.1)	NS ^5^
**Smoking**			
Never smoker (ref: ever smoker)	56	0.85 (0.48–1.5)	NIM
Current smoker (ref: no current smoker)	57	0.95 (0.42–2.1)	NS
**Alcohol use >7/week**			NS
No	60	ref	NS
Yes	51	0.70 (0.35–1.4)	
**Drug use**			
Yes	63	ref	NS
No	58	0.80 (0.3–2.1)	
**Comorbidities**			
Charlson comorbidity index	NA	0.87 (0.68–1.1)	NS
Normal kidney function	61	**ref**	NS
Impaired kidney function (eGFR ^6^ < 60)	39	**0.42 (0.17–1.0)**	
Crohn’s disease	63	1.3 (0.66–2.6)	NS
Ulcerative colitis	69	1.7 (0.73–4.0)	NS
Rheumatoid arthritis	57	0.95 (0.42–2.1)	NS
Psoriasis/psoriatic arthritis	48	0.99 (0.46–2.2)	NS
Solid organ transplant recipient	42	0.48 (0.21–1.1)	**0.40 (0.17–0.97)**
Time since organ transplantation	NA	0.96 (0.98–1.00)	NIM
≤12 months	60	Ref	
>12 months	38	0.41 (0.01–3.01)	
**Number of drugs at baseline**			
No drugs	74	**ref**	NS
One drug	58	0.49 (0.23–1.0)	
Two drugs	42	**0.26 (0.10–0.63)**	
Three drugs	50	0.36 (0.12–1.1)	
**cIM ^7^**	52	0.66 (0.37–1.1)	NS
**cIM monotherapy**	58	0.97 (0.50–1.9)	NIM
Prednisolone (>10 mg/day or 700 mg cumulative)	49	0.62 (0.29–1.3)	NS
Low-dose prednisolone	60	1.1 (0.40–3.2)	NS
Thiopurine	63	1.3 (0.54–2.9)	NS
Methotrexate	42	**0.37 (0.21–0.91)**	**0.37 (0.17–0.81)**
Calcineurin inhibitor	50	0.69 (0.28–1.7)	NS
Mycophenolate mofetil	42	0.57 (0.20–1.1)	NS
**bIM ^8^**	54	0.77 (0.43–1.4)	NS
**bIM monotherapy**	57	0.91 (0.47–1.8)	NIM
TNF-alpha inhibitor	59	1.04 (0.55–1.9)	NS
Other biological immunomodulators (non-TNF-alpha inhibitor)	45	0.55 (0.22–1.4)	NS
Ustekinumab (anti IL-12/23)	43	0.52 (0.11–2.4)	NS
Rituximab/ocrelizumab (anti-CD20)	6.0	0.14 (0.2–1.2)	**0.01 (0.01–0.81)**
**Nonsystemic agents (combined)**	70	1.8 (0.65–4.8)	NS
Vedolizumab (α4β7-integrin)	70	1.7 (0.43–6.8)	NIM
5-aminosalicylates	68	1.5 (0.43–5.0)	NIM

^1^ CI = confidence interval. ^2^ NA = not applicable. ^3^ NIM = variable not included in multivariable analysis. ^4^ BMI = body mass index. ^5^ NS = nonsignificant and factor not included in final model after stepwise backwards selection. ^6^ eGFR = estimated glomerular filtration rate. ^7^ Conventional immunomodulator, alone or combined with other drugs. ^8^ Biological immunomodulator, alone or combined with other drugs. Key comparison group are in bold font.

## Data Availability

Data available with corresponding author upon reasonable request.

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
