# Peer review of "Immunogenicity of the 13-Valent Pneumococcal Conjugate Vaccine (PCV13) Followed by the 23-Valent Pneumococcal Polysaccharide Vaccine (PPSV23) in Adults with and without Immunosuppressive Therapy"

_vaccines, 2022, doi:10.3390/vaccines10050795_

Round 1
Reviewer 1 Report
Present study is a well written report on immunogenicty of pneumococcal vaccination in immunocompromised. There are quite some previous studies published on this topic but the follow-up data on weaning immunity are less well described before.
Major comments
- in line with the above, I would recommend authors to describe the follow-up data and weaning immunity a bit more extensively and compromise some tables, ie in baseline characterics immunosuppressants are linked to underlying condition
- To conclusion to improve the vaccine response by an earlier booster is rather specultative and a topic of discussion. The slope of the curve in immunocompromised is similar to the heathly controles but the primary response is lower,. This can also be an argument to give additional PCV vaccinations. Even more since titre level chosen as ‘protective’ can be variable and is arbitrary, as no level of ‘seroprotection’ against most pneumococcal disease has been established.
- Several studies (ie 35317410) on vaccine immunogenity in immunocompromised show that B-cel depletion is associated more impaired responses compared to antiTNF, therefore I would recommend to separate these groups in complete analysis and not only in multivariate model
Minor comments
- Introduction, please check for important previously publised studies on pneumococcal vaccin immunogenicy in immunocompromised and refer to these studies in the introduction and not only in the discussion (see 32530360 and ie ...is missing
- Reference for cut off value of adequate response of pneumococcal test is missing in method section
Author Response
Dear reviewer,
Thank you for your thorough review, which helped us improve our manuscript.
Find attached our point-to-point response to your concerns.
Kind regards on behalf of all co-authors,
Hannah Garcia Garrido

Reviewer 2 Report
This paper reports a prospective cohort study in 233 adult patients conducted in the Netherlands between 2018 and 2021 aiming to investigate the immunogenicity of the 13-valent pneumococcal conjugate vac-27 cine (PCV13) + 23-valent pneumococcal polysaccharide vaccine (PPSV23) in immunocompromised patients. More specifically, it focuses on patients receiving a panel of immunosuppressive drugs and compare them to controls (not receiving any immunosuppressive therapy).
Overall this paper is well written, the scientific rationale for the study is clear, and both the methods and results are clearly laid out. I think it should be published with only minor editing.
I have noted that the control group has >50% with IBD and some with other autoimmune condition, could that have had an impact of the outcome of the study? (e.g., underestimating the estimation of the difference in seroprotection rates between groups).
Author Response
Dear Reviewer,
Thank you for reviewing our manuscript and your kind words.
Find attached our answer to your query.
Kind regards on behalf of al co-authors,
Hannah Garcia Garrido
